# Revealing the predictability of intrinsic structure in complex networks

Jiachen Sun[1,2], Ling Feng [3,4], Jiarong Xie[1], Xiao Ma[1], Dashun Wang[5,6,7] & Yanqing Hu[1,8]*

Structure prediction is an important and widely studied problem in network science and machine learning, finding its applications in various fields. Despite the significant progress in prediction algorithms, the fundamental predictability of structures remains unclear, as networks' complex underlying formation dynamics are usually unobserved or difficult to describe. As such, there has been a lack of theoretical guidance on the practical development of algorithms for their absolute performances. Here, for the first time, we find that the normalized shortest compression length of a network structure can directly assess the structure predictability. Specifically, shorter binary string length from compression leads to higher structure predictability. We also analytically derive the origin of this linear relationship in artificial random networks. In addition, our finding leads to analytical results quantifying maximum prediction accuracy, and allows the estimation of the network dataset potential values through the size of the compressed network data file.

[1] School of Data and Computer Science, Sun Yat-sen University, Guangzhou 510006, China. [2] School of Electronics and Information Technology, Sun Yat-sen University, Guangzhou 510006, China. [3] Institute of High Performance Computing, A*STAR, Singapore 138632, Singapore. [4] Department of Physics, National University of Singapore, Singapore 117551, Singapore. [5] Center for Science of Science and Innovation, Northwestern University, Evanston, IL, USA. [6] Kellogg School of Management, Northwestern University, Evanston, IL, USA. [7] McCormick School of Engineering, Northwestern University, Evanston, IL, USA. [8] Southern Marine Science and Engineering Guangdong Laboratory, Zhuhai 519082, China. *email: huyanq@mail.sysu.edu.cn

Predicting structure or links of network is commonly defined as estimating the likelihood of existence of unobserved links or potential new links[1–5]. Network as a common form of data representation is ubiquitous across a broad range of fields from biology[6–8], recommendation systems[9,10] to social media[1,3,11]. It represents the complex relationships or interactions among the elements in a system, which usually cannot be well described by simple mathematics. Hence, machine learning has been widely used in link predictions[10–18]. For instance, predicting protein–protein[7,8] or drug–target[19–21] interactions can guide more accurate biological experiments and reduce the experimental costs and time[2,6]. Despite the immense ongoing efforts in developing prediction algorithms, the fundamental understanding of the intrinsic prediction limit that can provide the much needed guidance is lacking. The difficulty lies with the fact that it is almost impossible to know the exact underlying mechanism of the network formation. In addition, the real networks are usually highly complex and gigantic in size, with many short feedback loops which cannot be effectively analyzed—a challenge[22] faced by both statistical physicists and computer scientists. Hence, understanding the prediction limits and quantifying it remains to be a long-standing challenge[2,23].

In this study, we reveal the intrinsic predictability of networks through their structural properties. Intuitively, a network structure that can be captured with a few words means it is simple, and its links are easily predictable such as one-dimensional chain and two-dimensional lattice. Conversely, if a network requires lengthy description, then it has very complicated structure and its links are hard to predict. In computer language, the structure of any network can be encoded into binary strings. This motivates us to find the underlying relationship between the network compression length of the shortest binary string and the prediction limit. On the other hand, the inherent prediction limit is the maximum predictability or performance that the theoretical best predicting algorithm (TBPA) can achieve. Here the network structure predictability is defined as the performance of TBPA and is quantitatively measured using entropy in this paper. Thus, without knowing the exact underlying dynamics of network formation which determines this limit, we can use the best prediction algorithm available (BPAA)[1,2,23,24] to approximate the performance of TBPA. With these two quantities of the shortest lossless compression length and performance of TBPA, here, for the first time, we discover a linear relationship between them in different empirical networks such as biological networks, social networks, and technology networks (see Supplementary Note 1 for the detailed description of studied networks). Our finding implies that the shortest compression length of a network can tell us the structure predictability, which sets the limit of any prediction algorithm.

## Results

**Network shortest compression length.** The shortest possible compression length can be calculated by a lossless compression algorithm[25], which is a proven optimal compression for random networks and efficient for many real networks. Since all of the structural properties of two isomorphic networks are exactly the same, the algorithm first encodes the network structure into a string of binary codes, from which an isomorphic network can be reconstructed. To further remove the correlations in the structure, the binary string is compressed by a recursive arithmetic encoder[26], which exploits the dependencies between the symbols in this binary string. After these two compression operations (see Fig. 1a), the length of the final bit string should be close to the network's structural entropy and can well measure its randomness (Shannon's source coding theorem[27], see Supplementary

Note 2). The length of this bit string is expected to increase as the structure becomes more random, and has been validated in Fig. 1b, which shows shuffled networks has longer compression length compared with the original empirical networks. As randomness in shuffling increases, the compression length increases monotonically.

Naturally the compression length is longer for a network with more nodes and links given the same randomness, i.e., the size of the network contributes to the compression length, rather than just the level of randomness of the network itself. In order to remove the size effect of the network, we normalize the compression length $L$ through dividing it by the theoretical maximum compression length $\mathcal{R}$, which corresponds to Erdős–Rényi (ER)[25,28] network of the same number of nodes and links (see Supplementary Note 2). The normalized value of the shortest compression length $L^*$ is given by

$$L^* = \frac{L}{\mathcal{R}}. \tag{1}$$

Here $\mathcal{R} = \binom{N}{2}h(q) - N\log N$[25], where $N$ is the number of nodes and $q$ is the probability having a link between any pair of nodes, also expressed as $q = \frac{E}{\binom{N}{2}}$, in which $E$ is the number of edges. $h(q)$ is the binary entropy given by $h(q) = -q\log q - (1-q)\log(1-q)$. Here log denotes the logarithmic operation with base 2, and we use this convention throughout this work.

**Structure predictability measured by algorithm performance.** Now we compare the shortest compression length with the performance of BPAA which approximates the structure predictability of network. In a large literature on link prediction algorithm[1–6,11,15,16], it is common to assign each unlinked pair of nodes (possible missing link) a score, and higher score means higher likelihood of this pair being a missing link. Here we adopt the leave-one-out[31,32] approach and use the score of the existing links to quantify the BPAA performance. First we remove a link $e_i$, and then use one particular prediction algorithm to estimate a score for each of the unlinked node pairs including $e_i$. Based on the ranking of the scores in descending order, we get the ranking $r_i$ of the removed link $e_i$. So when $r_i = 1$, it means that the algorithm tells us the removed link is the most probable missing link among all other unlinked node pairs. We carry out this calculation of $r_i$ from the original network for every link at a time, and obtain a sequence of rank positions $\mathbf{D} = \{r_1, r_2, \ldots, r_E\}$, where $E$ is the total number of links in the original network.

Naturally, the entropy of the distribution of $\mathbf{D}$ is a good holistic measure of the algorithm performance. For example, an ideal algorithm for a highly predictable network would have $\mathbf{D} = \{r_1 = r_2 = \cdots = r_E = 1\}$, thus yielding the lowest distribution entropy of $r_i$. Conversely, an ideal algorithm for a network with low predictability has very different values in $\mathbf{D}$, leading to a high entropy of its distribution. In our calculation of this algorithm performance entropy $H$, the value $r_i$ can vary in the range $1 \leq r_i \leq \frac{N(N-1)}{2} - \frac{\langle k \rangle N}{2} + 1 \approx \frac{N^2}{2}$, where $\frac{N(N-1)}{2} - \frac{\langle k \rangle N}{2} + 1$ is the total number of unlinked pairs, and $\langle k \rangle$ is the network's average degree. Thus we divide such range into bins with equal width $N$ to avoid the contribution of network size $N$ on the result (see Supplementary Note 3 for the discussion of other bin widths), and calculate $H$ based on the probability distributions of the $N/2$ bins: $H = -\sum_{j=1}^{N/2} p_j \log p_j$ where $p_j$ is the probability of $r_i$ in bin $j$ (Supplementary Note 3). Figure 1c illustrates an example of such distribution for the Metabolic network[29] by RA algorithm[30].

It is worth mentioning that using rank distribution entropy to measure the algorithm performance is the innovation in our work. In particular, we implement the leave-one-out method[31,32]

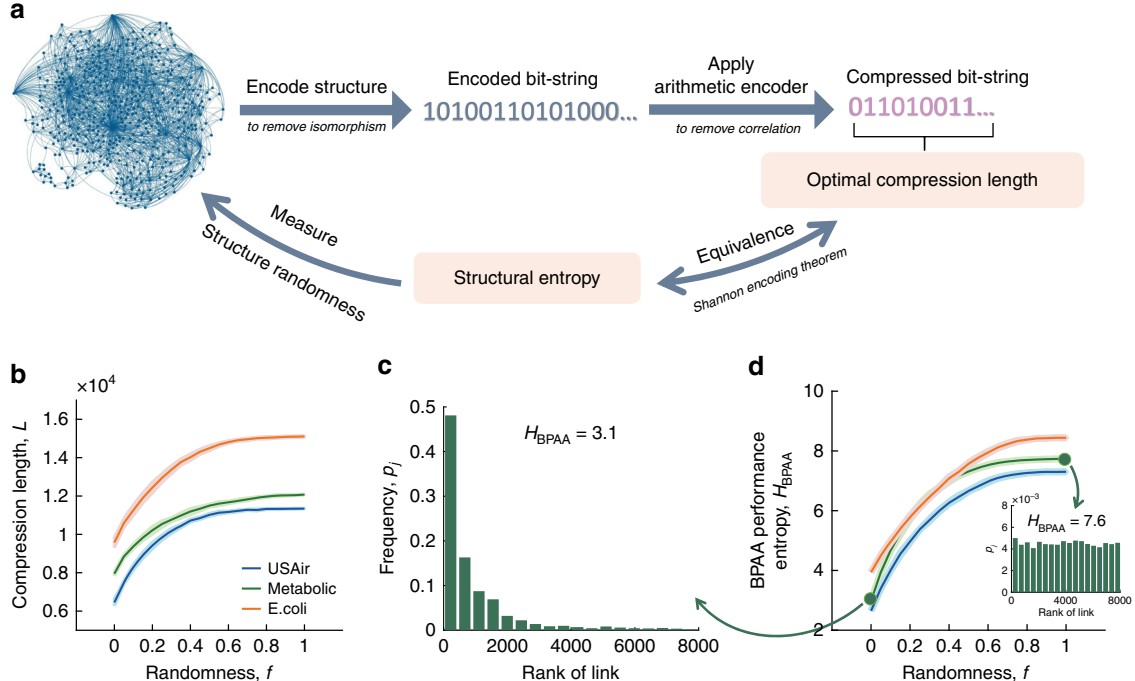

**Fig. 1 Network compression and BPAA performance entropy. a** Illustration of compression on network structural information. The lossless compression algorithm converts the network structure into a binary string, without retaining the labels of the nodes, and then compress the bit string to achieve the shortest compression length. **b** The shortest compression length of the shuffled networks. Here the shuffling operation is that we randomly pick $f$ fraction of links from the original network and rewire them randomly. The compression length increases monotonically with $f$ until $f = 1$, when the network becomes ER network. **c** Distribution of $r_i$ of Metabolic network[29]. Here we use RA algorithm[30] to do the prediction and each bar has the width equal to the network size $N$. The entropy of this distribution is representing the overall predictability. **d** The BPAA performance entropy $H_{BPAA}$ vs. the link shuffling fraction $f$. We can see a monotonic trend similar to (**b**) is present for all of the three networks. Each line is the average of 50 simulations and the shaded region denotes the standard deviation. (Inset) Distribution of $r_i$ of ER network with the same number of nodes and links as Metabolic network. The value of $p_j$ is the average of 50 simulations.

to obtain the rank distribution. The other ranking methods[2,3,23] which pre-remove $x$ fraction of links ($x$ is usually 5% or 10%) with multiple samples are similar to the leave-one-out method, and would yield similar results as leave-one-out if $x$ is very close to 0 (see Supplementary Note 4 and Supplementary Fig. 6). Using the leave-one-out method, there are three main advantages. Firstly, it is parameter-free, i.e., we do not need to consider the impact of the choice of $x$, and allows analytical study. Secondly, doing so preserves the original network structure as much as possible, such that the intrinsic true predictability is also preserved. Thirdly, the leave-one-out method has negligibly small fluctuation (see Supplementary Fig. 8). The main drawback is the higher computational complexity compared with only using 1 sample of removing $x$ fraction of links. However, if multiple samples of removing $x$ fractions are considered, the advantage of complexity of above other methods will not be significant over ours.

In order to obtain a good approximation of the TBPA performance of the network that is less dependent on a particular prediction algorithm being used, we employ a range of widely applied 11 prediction algorithms (see Supplementary Note 5), such as structural perturbation method (SPM)[23], local random walk (LRW)[33], average commute time[24] and common neighbor[1] to calculate the ranking positions $r_i$, and use the one that gives the lowest $H$ value $H_{BPAA}$ (Table 1) as the closest estimate of the network's predictability $H_{TBPA}$. To further explore the relationship between network structure and $H_{BPAA}$, it can be seen in Fig. 1d that as the network structure is progressively randomized due to shuffling, the entropy $H_{BPAA}$ increases, signifying a decrease in predictability. Eventually with enough shuffling, the

algorithm performance entropy $H$ approaches that of ER networks with a nearly uniform distribution of **D** (Fig. 1d inset). To further remove the effect of network size $N$ and average degree $\langle k \rangle$, we normalize the BPAA performance entropy $H_{BPAA}$ value by $\log N - 1$, which is the BPAA entropy of an ER network of the same size and average degree. Hence the normalized BPAA performance entropy $H_{BPAA}$ is defined as

$$H_{BPAA}^* = \frac{H_{BPAA}}{\log N - 1}. \qquad (2)$$

**Empirical linear relationship**. Having both network's structural shortest compression length and the BPAA performance, we are able to find the relationship between the two. Surprisingly, we discover a clear linear relationship between the $H_{BPAA}^*$ and $L^*$ across 12 empirical networks in very different fields including biology, social media, transport and economics as shown in Fig. 2a:

$$H_{BPAA}^* \approx 1.63 L^* - 0.63. \qquad (3)$$

It is worth mentioning that, the way $H$ and $L$ are normalized is critical to obtain the linear relationship, as without proper normalization, the network's compression length cannot serve as a good indicator to quantify the structure predictability of a network (see Supplementary Note 6).

We find that such a linear relationship in Eq. (3) still exists even after we shuffle links as seen in Fig. 2b, or randomly add/ delete links as seen in Fig. 2c, d. This implies that the linear relationship is universal and invariant under perturbations. Besides, such invariance can be validated through the additive

**Table 1 Normalized algorithm performance entropy.**

| Network | $N$ | $\langle k \rangle$ | Normalized Algorithm Performance Entropy | | | | | | | | | | | |
|---|---|---|---|---|---|---|---|---|---|---|---|---|---|---|
| | | | CN | L3 | Jaccard | Salton | PA | AA | RA | ACT | LRW | SRW | SPM | BPAA |
| *USAir* | 332 | 12.80 | 0.491 | 0.504 | 0.669 | 0.650 | 0.585 | 0.448 | 0.363 | 0.578 | 0.447 | 0.438 | 0.419 | **0.363** |
| *C.elegans* | 297 | 14.46 | 0.748 | 0.746 | 0.845 | 0.849 | 0.898 | 0.732 | 0.723 | 0.905 | 0.664 | 0.708 | 0.665 | **0.664** |
| *Metabolic* | 453 | 8.94 | 0.601 | 0.614 | 0.773 | 0.755 | 0.753 | 0.453 | 0.385 | 0.831 | 0.455 | 0.413 | 0.419 | **0.385** |
| *PPI* | 2617 | 9.06 | 0.488 | 0.430 | 0.527 | 0.532 | 0.723 | 0.460 | 0.443 | 0.674 | 0.350 | 0.404 | 0.254 | **0.254** |
| *E.coli* | 761 | 5.98 | 0.516 | 0.534 | 0.589 | 0.599 | 0.656 | 0.472 | 0.466 | 0.613 | 0.435 | 0.444 | 0.472 | **0.435** |
| *Political blogs* | 1224 | 27.30 | 0.680 | 0.650 | 0.795 | 0.798 | 0.767 | 0.674 | 0.673 | 0.779 | 0.664 | 0.702 | 0.617 | **0.617** |
| *Facebook* | 786 | 35.68 | 0.561 | 0.620 | 0.512 | 0.507 | 0.880 | 0.544 | 0.493 | 0.760 | 0.552 | 0.550 | 0.465 | **0.465** |
| *Bio-CE-GT* | 924 | 7.01 | 0.589 | 0.513 | 0.658 | 0.655 | 0.800 | 0.520 | 0.491 | 0.797 | 0.394 | 0.428 | 0.416 | **0.394** |
| *Soc-wiki-vote* | 889 | 6.56 | 0.735 | 0.713 | 0.801 | 0.803 | 0.818 | 0.729 | 0.739 | 0.814 | 0.687 | 0.693 | 0.717 | **0.687** |
| *Econ-wm1* | 277 | 17.19 | 0.487 | 0.372 | 0.631 | 0.609 | 0.482 | 0.480 | 0.505 | 0.638 | 0.400 | 0.531 | 0.260 | **0.260** |
| *Econ-mahindas* | 1258 | 11.93 | 0.724 | 0.503 | 0.780 | 0.752 | 0.629 | 0.725 | 0.716 | 0.712 | 0.557 | 0.660 | 0.471 | **0.471** |
| *Tech-routers* | 2113 | 6.27 | 0.576 | 0.574 | 0.651 | 0.648 | 0.751 | 0.548 | 0.559 | 0.756 | 0.483 | 0.503 | 0.481 | **0.481** |

The algorithm performance entropy $H$ is normalized by the BPAA entropy of an ER network of the same size and average degree for each individual network. The lowest value among the 11 algorithms is highlighted in boxes, and is used to approximate the $H^*_{\text{TPAA}}$ for each network as the boldface number in the BPAA column.

property of the shortest compression length[25,27], i.e., the sum of the shortest compression lengths $L_A$, $L_B$ of two independent networks $A$ and $B$ (with the same nodes but different links) is the same as the compression length $L_{A+B}$ of the combined network $A + B$ (Fig. 2e). This is also valid for the compression algorithm in case of shuffling and adding links at random on empirical networks (Fig. 2f, g).

We now illustrate how to use the linear relationship in Eq. (3) to quantify the performance of a prediction algorithm. By comparing the Jaccard algorithm[35] performance entropy with the linear relationship (see Supplementary Fig. 11), it can be seen that for a given network, the further is the performance entropy from the linear line, the less optimal it is from the best possible prediction results (see Supplementary Note 7). This quantifiable distance can serve as a benchmark to practical algorithms in a given network. In other words, the further it is from the linear line, the more potential improvement it is possible for any new algorithm to achieve, warranting further effort on algorithm development. Conversely, if the algorithm performance entropy lies close to the line, it means that the algorithm is already performing quite well, and further exploration on algorithms is unlikely to yield significant improvement, suggesting diminishing returns in developing new algorithms.

Since the compression process can be executed through different node sequences, we also study the effect of compressed node sequences on the compression length, such as random, high degree priority and low degree priority. We find that the differences in compression lengths are very small (see Supplementary Fig. 2). Hence, we can see this compression algorithm is reliable when operating on real networks. Naturally, such relationship between $H^*_{\text{BPAA}}$ and $L^*$ indicates that the link prediction limit of a network can be directly inferred from the network topological structures. More explicitly, by compressing the network structure into binary string and calculate the string's length, one can estimate the link prediction limit $H^*_{\text{BPAA}}$. Apart from the theoretical significance of the relation, one practical advantage of estimating $H^*_{\text{BPAA}}$ with $L^*$ is that $L^*$ is only dependent on the network topological structure and has low computational complexity $O(N + E)$[25]. In contrast, estimation of $H^*_{\text{BPAA}}$ requires a large number of available prediction algorithms, which usually have high computational complexity. Specifically, for calculating the rank distribution entropy $H$, we need to

employ an algorithm for every link in a network to obtain its ranking. Consequently, the computational complexities of algorithms like SPM[23] and LRW[33] are about $O(N^3 E)$ and $O(N\langle k\rangle^n E)$ in estimating $H$, where $n$ is an integer constant. Furthermore, an additional benefit of the compression length is that it can be served as an independent indicator to identify missing or false links (see Supplementary Note 8).

**Theoretical linear relationship.** The empirical linear relationship inspires us to further figure out the underlying mathematical connection between the network shortest compression length and link prediction limit. For simplicity, we assume that an artificial network is generated from a static random matrix $Q$ whose entry $q_{ij}$ denotes the link formation probability between node $i$ and $j$. According to Shannon's source coding theorem[27], the shortest compression length $L$ of this artificial network is $L = \sum_{i>j} h(q_{ij}) - N\log N$, which is the structural entropy[25], where $h(q_{ij})$ is the binary entropy given by $h(q_{ij}) = -q_{ij}\log q_{ij} - (1 - q_{ij})\log(1 - q_{ij})$. To simplify representation, we use $U$ to denote the term $-\sum_{i>j} q_{ij}\log q_{ij}$ and expand the logarithm as a Taylor series, of which the higher-order small terms can be neglected, yielding:

$$L \approx U - N\log N + \frac{N\langle k\rangle}{2\ln 2}, \qquad (4)$$

where ln denotes the natural logarithm, and we use this convention throughout this work. Since the occurrence of each link in this artificial network solely depends on the $q_{ij}$, it is certainly true that the most accurate score that a TBPA could achieve should be exactly equal or proportional to $q_{ij}$. Thus, without employing any prediction algorithm, we can obtain the ranking sequence $D$ (see the right part of Fig. 3a) directly from $Q$ and are able to numerically quantify $H^*_{\text{TBPA}}$ in the same way that we calculate $H^*_{\text{BPAA}}$.

We find that $H^*_{\text{TBPA}}$ can be theoretically estimated by $Q$, given that the TBPA ranking distribution clearly agrees well with the distribution of $q_{ij}$ in $Q$ (Fig. 3b). The entropy of the latter is given by $H_Q = -\sum_{i>j} \frac{q_{ij}}{\sum_{i>j} q_{ij}} \log \frac{q_{ij}}{\sum_{i>j} q_{ij}}$. Recalling that in the calculation of $H_{\text{BPAA}}$, we have divided the ranking range into bins with equal width $N$. In such case, this coarse-grained distribution of $Q$ is done by replacing every $N$ values of $q_{ij}$ (in the descending order) by their average value $\tilde{q}_{ij}$ (see the left part of Fig. 3a), and the

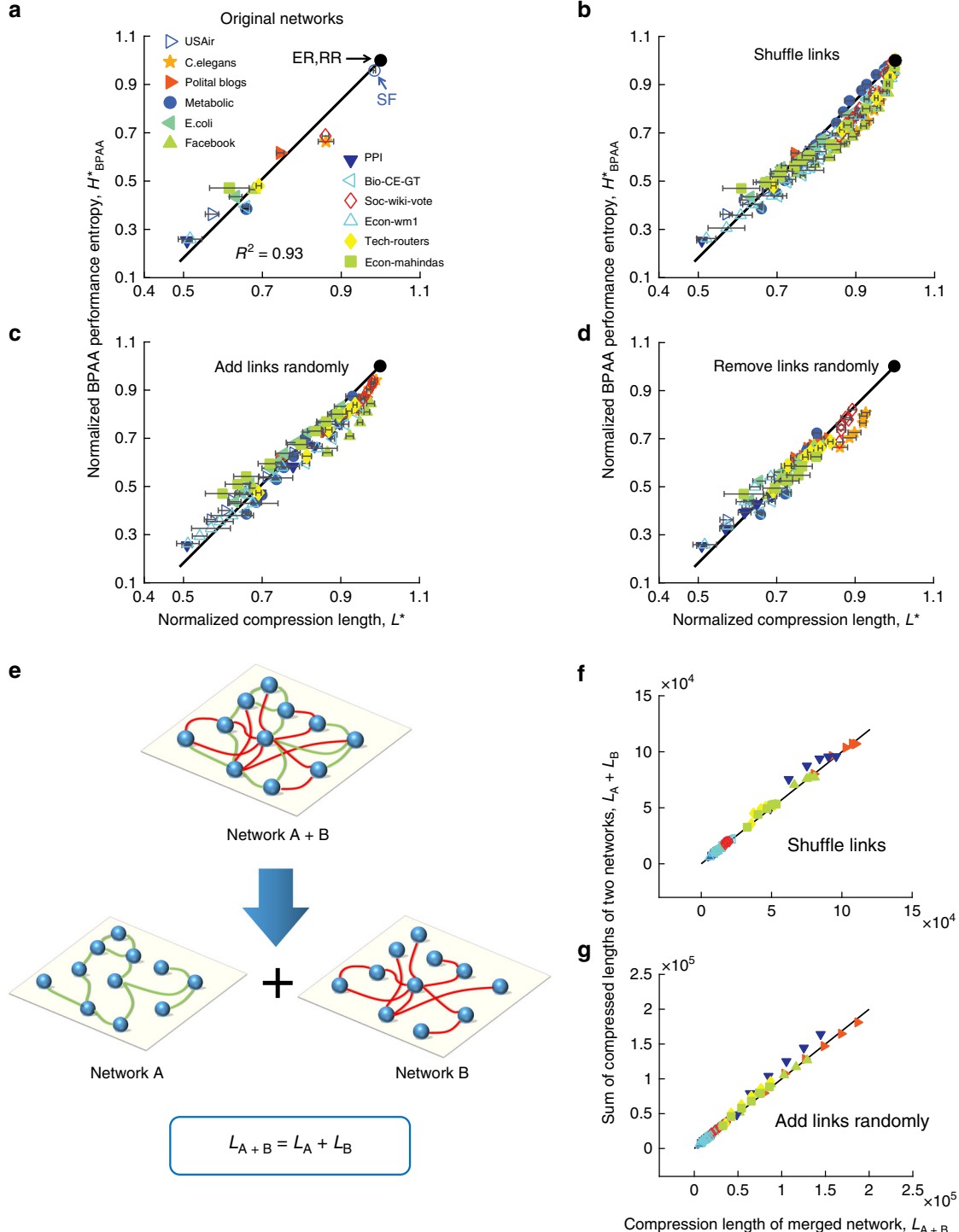

**Fig. 2 Network prediction limit vs. shortest compression length on empirical networks. a** For empirical networks drawn from different fields, the values $(L^*, H^*_{BPAA})$ fall on the linear relationship in the black straight line given by Eq. (3). The horizontal error bars indicate the standard derivation of the compression length across 50 calculations (see Supplementary Note 2). The standard deviation of the slope is 0.06 obtained by the bootstrap method[34] and the coefficient of determination $R^2$ is 0.93. **b** Similar plot to (**a**), with each empirical network having its links shuffled different number of times up to all of original links. After shuffling, the values $(L^*, H^*_{BPAA})$ move towards the point of ER network but still fall on the same linear straight line as in **a**. **c** Random addition of links up to the same number of links of the original network and (**d**) random removal of links up to half of original number of links in the various empirical networks. The resulting compression length $L^*$ and predictability $H^*_{BPAA}$ still have the same linear relationship as the original networks in a A. **e** A schematic representation of combing two networks with the same nodes into one. **f** Values of $L_{A+B}$ vs. $L_A + L_B$ in the case of shuffling links in empirical networks. A network after shuffling operation can be viewed as consisting of two independent subnetworks **A** and **B**: **A** refers to a network which only contains the shuffled links, i.e., an exact ER network; and the **B** is the remaining network with original links in the empirical network. **g** Similar plot to (**f**) in the case of randomly adding links in empirical networks. Here network **A** is an ER network containing all of the randomly added links while network **B** refers to the original real network.

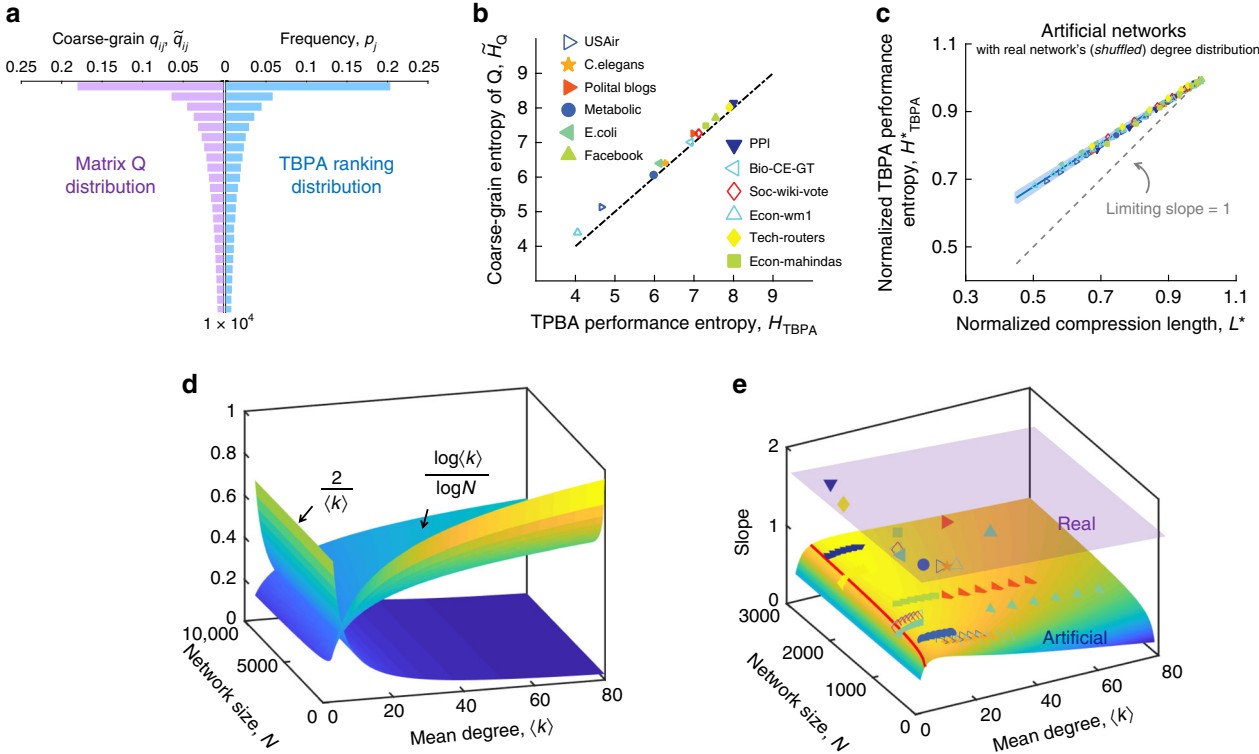

**Fig. 3 Theoretical relationship between the shortest compression length and network prediction limit.** Networks are generated from a static random matrix **Q** with elements generated from the degree distribution of empirical networks. **a** Comparison between the ranking distribution from TBPA and that from random matrix **Q** based on Metabolic network[29]. The left half represents the ranking distribution of coarse-grained probabilities $\tilde{q}_{ij}$, each of which is the average value of every $N$ values of $q_{ij}$ ranked in a descending order. The right half illustrates the TBPA ranking distribution of existing links for this artificially generated network. The probability $p_j$ is an average value from 100 simulations. **b** Values of coarse-grained entropy $\tilde{H}_\mathbf{Q}$ from the probability distribution $\tilde{q}_{ij}$ vs. TBPA performance entropy $H_{\text{TBPA}}$ for all of the artificial networks generated from the empirical ones. Each dot corresponds to the value of $(H_{\text{TBPA}}, \tilde{H}_\mathbf{Q})$ of an artificial network, showing a good match between the two values. **c** Theoretical linear relationship between $L^*$ and $H^*_{\text{TBPA}}$ calculated based on the empirical networks' degree distributions. Points of the same shape and color correspond to an empirical network (bottom-left) and its different shuffled versions, i.e., having its links shuffled different number of times up to all of original links. Each point represents the value pair of $(L^*, H^*_{\text{TBPA}})$ of an artificial network with the same degree distribution as a corresponding (shuffled) empirical network. $L^*$ is calculated from Eq. (1) and $H^*_{\text{TBPA}}$ is calculated from Eq. (6). The blue solid line is the average of 12 studied networks' analytical slopes obtained from Eq. (7) and the shaded region denotes the standard deviation. The gray dash line is the analytical result of Eq. (9) for the limit of $\langle k \rangle \to \infty$. **d** Competition between two terms of the slope in Eq. (8). **e** The empirical and theoretical value of slope in the linear relationship between $L^*$ and $H^*_{\text{TBPA}}$. The purple plane denotes the plane of empirical value at 1.63, and each colored point on the plane represents a real network. The lower curved surface below represents the theoretical values of the slope given by Eq. (7), and each colored point on the lower surface represents an artificial network constructed with the same degree distribution as original empirical networks, and the subsequent points with the same color are the artificial networks with randomly added links to increase $\langle k \rangle$. The red curve $(\ln N = \langle k \rangle)$ on the lower surface is an estimated boundary (see Supplementary Note 9) that our theories are not valid far from the left side of it.

resulting entropy satisfies $\tilde{H}_\mathbf{Q} = H_\mathbf{Q} - \log N$ (see Supplementary Note 9). The difference between $H_{\text{TBPA}}$ and $\tilde{H}_\mathbf{Q}$ is due to the fact that for $H_{\text{TBPA}}$, link prediction algorithms only measure the likelihood of unobserved pairs of nodes forming a link. Such difference, however, contributes negligibly to the calculation of entropy as shown in Fig. 3b, yielding $H_{\text{TBPA}} \approx \tilde{H}_\mathbf{Q}$. Taken together, we obtain (see details in Supplementary Note 9):

$$H_{\text{TBPA}} \approx \frac{2}{N\langle k \rangle}U + \log\frac{\langle k \rangle}{2}. \quad (5)$$

We then normalize $H_{\text{TBPA}}$ by $\log N - 1$ through the same way of Eq. (2), yielding

$$H^*_{\text{TBPA}} = \frac{H_{\text{TBPA}}}{\log N - 1}. \quad (6)$$

Combining with Eqns. (1), (4)–(6) and eliminating the variable $U$,

we obtain the linear relationship between $L^*$ and $H^*_{\text{TBPA}}$:

$$H^*_{\text{TBPA}} \approx \frac{2[\binom{N}{2}h(q) - N\log N]}{\langle k \rangle N\log\frac{N}{2}}L^* + \frac{\frac{2\log N}{\langle k \rangle} + \log\frac{\langle k \rangle}{2} - \frac{1}{\ln 2}}{\log\frac{N}{2}}. \quad (7)$$

When $N$ is large Eq. (7) simplifies to

$$H^*_{\text{TBPA}} \approx (1 - \frac{2}{\langle k \rangle} - \frac{\log\langle k \rangle}{\log N})L^* + \frac{2}{\langle k \rangle} + \frac{\log\langle k \rangle}{\log N}. \quad (8)$$

In the thermodynamics limit Eq. (8) can be further approximated when $\log N \gg \log\langle k \rangle$:

$$H^*_{\text{TBPA}} \approx (1 - \frac{2}{\langle k \rangle})L^* + \frac{2}{\langle k \rangle}. \quad (9)$$

Usually for many real networks, $\frac{\log\langle k \rangle}{\log N}$ is not negligible, and Eq. (8) is a better approximation. The detailed mathematics above is given in Supplementary Note 9.

To validate Eq. (7), we first construct artificial networks based on each empirical network in Fig. 2 with the same degree sequence, but a simple network formation mechanism based that

only depends on the static probability distribution matrix $\mathbf{Q}$, with each element $q_{ij} = \frac{k_i k_j}{2E}$, where $k_i$ denotes the degree of node $i$ in an empirical network. Secondly, we shuffle each above network by randomly picking a fraction of links and rewiring them randomly (same operation in Fig. 2b) and then construct artificial networks based on each shuffled empirical network's degree sequence. In the shuffling process, the randomness of the network increases with the increasing of the shuffling strength. Theoretically, the values $(H^*_{\mathrm{TBPA}}, L^*)$ changes along the straight line predicted by Eq. (7). It can be seen from Fig. 3c that this theoretical relation in Eq. (7) is well captured by our simulation for the different networks and their shuffled versions. We have also considered two other genuine edge-independent synthetic networks: the degree-correlated stochastic block model[36] characterizing the community structure, and the latent-geometric network model[37] characterizing network spatial structure. Both models have been shown to reproduce certain structural properties of real network. We find that the structure predictability properties obtained by these two models also follow our analytical result for artificial networks (see Supplementary Note 9 and Supplementary Fig. 15).

Surprisingly, the artificial networks generated by matrix $\mathbf{Q}$ leads to $(L^*, H^*_{\mathrm{BPAA}})$ pairs falling on a straight line as seen in Fig. 3c, with a slope that is different from the one given by Eq. (9) from the thermodynamics limit approximation. A careful examination shows that the artificial networks' average degrees $\langle k \rangle$ increase with the network sizes $N$ faster than $\log N$, balancing out the changes in both $\frac{\log\langle k \rangle}{\log N}$ and $\frac{2}{\langle k \rangle}$ in Eq. (8) (see Fig. 3d), therefore keeping the slope relatively constant around 0.7 instead of the thermodynamics limit 1 (see the plateau in Fig. 3e). For the approximation result in Eq. (9), we find that the thermodynamic limit slope 1 with large $\langle k \rangle$ can be realized in the artificial network with size larger than $10^{100}$ (see Supplementary Fig. 14). But this large number is almost impossible in real networks. Moreover, the values of the slope between $H^*_{\mathrm{TBPA}}$ and $L^*$ of artificial networks are significantly lower than the empirical slope of 1.63 (Fig. 3e). This may be due to the more complex mechanisms and constraints in empirical network formations compared with simplistic random link connection in the artificial networks. Intuitively one expects the structure predictability of the empirical network to be higher than that of the purely random due to such additional constraints or mechanisms, and it is reflected in higher slope observed. Such hypothesis does not directly address the differences quantitatively, yet it sheds some light on the complex relationship between network entropy and structure predictability. At the same time, the empirical value of 1.63 hints at some strong universal mechanism that is common to all of these empirical networks.

**Bounds of link prediction precision.** In many practical applications involving link prediction, like recommendation system, a prediction algorithm gives the most likely missing links in the network among all possible links. One way to assess the algorithm's prediction precision is through the success of prediction[1–6,11,15,16]. In other words, we can look at the probability $p_1$ that the distribution of ranks given by the algorithm on the removed links defined in $\mathbf{D}$. Note that if we take the top $N$ predicted links to calculate the standard precision, then it is equivalent to $p_1$ used in this paper, since $p_1$ refers to the fraction of correct links among the top $N$ predicted links. In practice, a good algorithm will give a ranking distribution that is usually monotonically decreasing, i.e., $p_1 \geq p_2 \geq \cdots \geq p_{N/2} \geq 0$ similar to Fig. 1b. That means the link predicted by the algorithm is indeed the most likely to be missing. Actually, we observe this phenomenon for all the algorithms and data used in this paper

(see Supplementary Fig. 26). Using the linear relationship between $L^*$ and $H^*_{\mathrm{BPAA}}$, for any network, we can calculate its normalized shortest compression length $L^*$ from its compressed binary string. Together with the assumption $p_1 \geq p_2 \geq \cdots \geq p_{N/2} \geq 0$, we arrive at an implicit function of the precision upper bound $\overline{p}_1$:

$$-\overline{p}_1 \log \overline{p}_1 - (1 - \overline{p}_1)\log\left(\frac{1 - \overline{p}_1}{N/2 - 1}\right) = (1.63L^* - 0.63)(\log N - 1).$$

$$(10)$$

The exact value of $\overline{p}_1$ can be obtained by solving Eq. (10) (see Fig. 4a). Additionally, the lower bound $\underline{p}_1$ of the prediction precision $p_1$ can be simply calculated through the following explicit formula (details in Methods section):

$$\underline{p}_1 \approx 2^{-(1.63L^* - 0.63)(\log N - 1)}. \qquad (11)$$

One can define the prediction precision in a more general way other than $p_1$. For instance, rather than defining the probability $p_1$ of the removed link falling into the first interval, one can loosen the definition to the first $C$ intervals ($1 \leq C \leq N/2$), i.e., defining precision as $P_C = \sum_{j=1}^{C} p_j$ (see Supplementary Note 10 for details on the derivations). Indeed, we validate the above upper bounds in the empirical networks as shown in Fig. 4b–d for USAir, Metabolic and E.coil network (see Supplementary Note 10 for the upper bounds for the other empirical networks).

**Commercial value of network dataset.** In practice, the prediction bounds enable something that was not possible before, for instance it can be used to estimate the commercial value of a network dataset, through its compressed size without developing any prediction algorithm. Thus, for a compressed network with $L$ bit, the conservative commercial value $V$ is

$$V \approx \Theta 2^{-(1.63L/\mathcal{R} - 0.63)(\log N - 1)}, \qquad (12)$$

where $\Theta$ is an external economic variable. For example, in the scenario of inferring interactions between proteins, $\Theta$ can be written as $\theta n$, where $\theta$ represents the unit experimental costs that can be saved if successfully predicting one interaction and $n(\leq N)$ denotes the number of predicted interactions. In other words, once we obtain the smallest number of bits $L$ needed to store a compressed network data file, we can directly derive the approximate value of this dataset through Eq. (12). Note that the commercial value we refer to above is the potential additional value that can be realized by predicting unobserved information from the network, without considering the overlapping value from external information outside the network structure. A more general framework of data commercial values when various external information is available is discussed in Supplementary Note 11.

## Discussion
Although we have seen that our finding applies to a broad range of empirical networks and their artificial counterparts, it is important to take note of certain limitations in practice. It is know from ref. [38] that Eq. (4) is accurate when $\ln N \ll \langle k \rangle \ll N - \ln N$ (see the red curve $\ln N = \langle k \rangle$ in Fig. 3d). Therefore when a network is extremely sparse or dense, the relationship between $L^*$ and $H^*_{\mathrm{BPAA}}$ may not be close to what we found. In addition, the network entropy estimation and structure predictability analysis assume randomness in structure. That means for regular networks like lattice structure, our finding does not hold. To illustrate the impact of regular structural features, we combine a regular network from a circular model network[39] (Fig. 5a) and real networks. The analysis on such synthetic networks show that

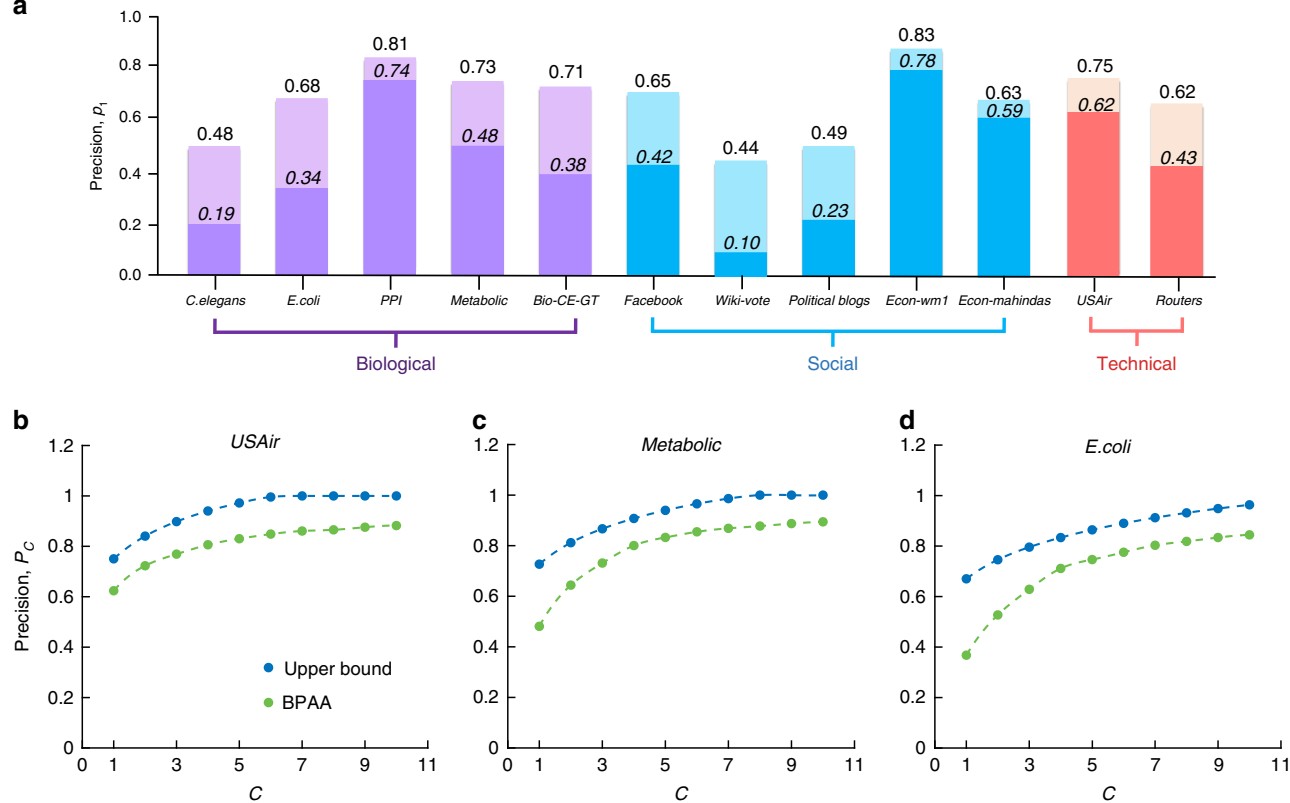

**Fig. 4 Upper bound of link prediction precision vs. BPAA's precision. a** For the various empirical networks (see Supplementary Note 1), the comparison of the precision $p_1$ of finding the correct missing link in the top ranking interval for the BPAA with the upper bound $\overline{p}_1$ obtained from Eq. (10). The light bars indicate the value of upper bound. The dark bars correspond to the BPAA's precision. (**b**), (**c**) and (**d**) are plotted for USAir, Metabolic and E. Coli networks on their BPAA's precision $P_C$ that the missing link is in the top $C$ intervals of rankings, compared with the upper bound $\overline{P}_C$ obtained from Eq. (17). Upper bounds on other real networks studied in this work are provided in Supplementary Note 10.

indeed, such regular structure in a random network makes the $H^*_{\text{BPAA}}$ vs. $L^*$ relation deviates from the slope of 1.63 as seen in Fig. 5b. Hence our result is more valid for networks without significant amount of regular links. That being said, if a network is very regular in structure, lattice for instance, its structure is easy to predicted and compressed due to high regularity involved.

In conclusion, for the first time we established a theoretical framework to quantify the intrinsic structure predictability in real and artificial random networks solely based on network structure, and independent from any prediction algorithms. With theoretical intuition on proper normalization, we uncover a universal linear relationship between the shortest compression length of the network structure and its structure predictability for a wide range of real networks such as biological networks, social networks and infrastructure networks, and analytically derived the mathematical origin of this linear relationship in artificial networks. In principle, such relationship can serve as a benchmark to quantify the performance of any practical prediction algorithm in non-regular networks. Leveraging upon this linear relationship, we can obtain the structure predictability that is intrinsic to the structure of complex networks and provide the accuracy bounds for any link prediction algorithm. In practice, our method can also be used to estimate the commercial value of a network dataset through its compressed length without using any link prediction algorithm. Our finding is demonstrated upon data structure in the form of networks. However, if the structural entropy and prediction limit are linked through the underlying dynamical process for data structures other than networks, it is likely that the predictability in some other machine learning problems for

different types of systems can also be inferred through similar approaches of optimal compression.

## Methods

**Compression algorithm**. Our compression scheme builds upon the seminal paper of[25], which is a two-step lossless compression of graphs. First we encode a network into two binary sequences $\mathbf{B}_1$, $\mathbf{B}_2$, through traversing all nodes on the network from an initial node, and encode nodes' neighbors according to specific rules[25]. In the second stage, both $\mathbf{B}_1$, $\mathbf{B}_2$ are compressed by an improved arithmetic encoder based on recursive splitting proposed by K. Skretting[26] to exploit the dependencies between the symbols of the sequence. After that we obtain two compressed binary sequences $\hat{\mathbf{B}}_1$, $\hat{\mathbf{B}}_2$, and we define the compression length of the network $L$ to be the total length of these two sequences:

$$L = \ell(\hat{\mathbf{B}}_1) + \ell(\hat{\mathbf{B}}_2), \tag{13}$$

where $\ell(\hat{\mathbf{B}}_1), \ell(\hat{\mathbf{B}}_2)$ are the length of $\hat{\mathbf{B}}_1$ and $\hat{\mathbf{B}}_2$, respectively. More details about the algorithm is provided in Supplementary Note 2.

**Upper and Lower Bounds of $p_1$ and $P_C$**. Here we demonstrate how to identify the upper and lower bounds of link prediction precision $p_1$ and $P_C$. The basic idea is to transform these problems into optimization problems of certain boundary conditions. Firstly, we calculate the network's BPAA performance entropy $H^*_{\text{BPAA}}$ through its shortest compression length $L^*$ given by Eq. (3). Its un-normalized value is $(1.63L^* - 0.63)(\log N - 1)$, and by the definition of entropy it can be written as

$$(1.63L^* - 0.63)(\log N - 1) = -\sum_{i=1}^{N/2} p_i \log p_i. \tag{14}$$

There are two constraints for $p_i$s. The first is that they sum up to unity, i.e.

$$\sum_{i=1}^{N/2} p_i = 1, \tag{15}$$

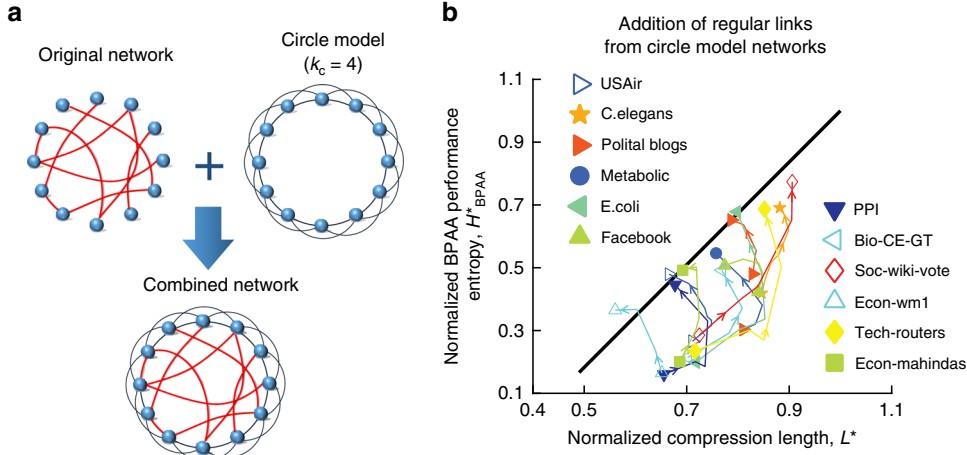

**Fig. 5 Effect of regular structures in networks. a** A schematic depiction of combining a circle model network (regular) into a real network. In the circle model network, each node has $k_c$ links with its closest nodes. **b** We decrease the regular links by using value of $k_c$ from 20 to 2 in the circular model network. Points with the same shape and color correspond to a real network. The arrow presents the flow of the value pair ($L^*$, $H^*_{BPAA}$) as $k_c$ decreases. It is clear that our theoretical prediction works better with less regular structural features.

and:

$$p_1 \geq p_2 \geq \cdots \geq p_{N/2} \geq 0, \quad (16)$$

as they are arranged in decreasing order.

With these boundary conditions, $p_1$ is maximized when the other probabilities are equal. Then it can be found that the upper bound of $p_1$ is given by Eq. (10).

And the minimum value of $p_1$ is when there are as many $p_i$s as close to $p_1$ as possible, leading to a lower bound given by Eq. (11).

For predictability of the top $C$ intervals $P_C = \sum_{j=1}^{C} p_j$, the upper bound corresponds to the case when the last $N/2 - C$ probabilities are the same, yielding

$$-\overline{P}_C \log \frac{\overline{P}_C}{C} - (1 - \overline{P}_C) \log\left(\frac{1 - \overline{P}_C}{N/2 - C}\right) = (1.63L^* - 0.63)(\log N - 1). \quad (17)$$

For the lower bound of $P_C$, the value of $p_C$ can be introduced to construct the minimization problem. It has the boundary condition of $0 \leq p_C \leq P_C/C$. The approximate lower bound is the solution of the following:

$$\underline{P}_C \approx \min\{P_C : -\gamma(P_C, p_C)\log\gamma(P_C, p_C) - (1 - \gamma(P_C, p_C))\log p_C \\ = (1.63L^* - 0.63)(\log N - 1)\}. \quad (18)$$

The more detailed mathematics is provided in Supplementary Note 10.

## Data availability

Data of this study is available at http://www.huyanqing.com/.

## Code availability

Source code of this study is available at http://www.huyanqing.com/.

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

## Acknowledgements
The authors would like to thank the two anonymous referees for the constructive suggestions and Shlomo Havlin, Bill Chi Ho Yeung, and Suihua Cai for the very help discussions. This work was supported by the National Natural Science Foundation of China under Grants No. 61773412, U1911201, U1711265, 61903385 and 61971454, the National Key R & D Program of China under grant 2018AAA0101200, Guangzhou Science and Technology Project, under Grant No. 201804010473, Guangdong Research and Development Program in Key Fields, under Grant No. 2019B020214002, the Chinese Fundamental Research Funds for the Central Universities Grant 19lgzd39, China Scholarship Council Program, under Grant No. 201906380135. D.W. was supported by the Air Force Office of Scientific Research under award number FA9550-19-1-0354.

## Author contributions
Y.H. conceived the project. J.S., L.F., D.W., and Y.H. designed the experiments. J.S. performed experiments and numerical modeling. J.S., L.F., J.X., X.M. and Y.H. discussed and analyzed the results. J.S., L.F. and Y.H. wrote the manuscript.

## Competing interests
The authors declare no competing interests.
