## [Peer Review File · Nature Communications]

Reviewers' comments:

Reviewer #1 (Remarks to the Author):

Report on the manuscript titled "Revealing the Predictability of Intrinsic Structure in Complex Networks" by J. Sun et al.

The authors report a linear relationship between the shortest compression length of a network and its link predictability. This result is established on a collection of real networks.

Authors support their empirical finding by deriving a linear relationship, albeit with a different slope for the class of edge-independent random graphs. Authors argue that the discovered linear relationship may serve as a tool for the estimation of commercial value of the network dataset.

The problem of link prediction and network reconstruction in general is an important problem in data sciences and is certainly worth investigation. The manuscript is fairly well-written and is accessible for non-specialists in the field.

1. My main concern is with respect to the importance of the reported finding. While the discovered linear relationship between the compression length and link predictability does not seem obvious, it is not at all clear what its immediate benefits are.

The authors claim that the discovered relationship may help evaluate the commercial value of the network dataset. While this might be true, one can argue that the compression length itself may serve as such indicator: small length implies that the dataset is regular and easy to predict, on the other hand a long compression length implies that the network dataset is more random and therefore is less predictable. I challenge the authors to think about better-defined applications of their finding.

Imagine, for instance, that you have an incomplete dataset where only a small fraction of links has been captured. Besides, some of the captured links are false-positives. Can the authors think about the way to estimate the completion cost of the dataset?

Can one use the compression length to estimate link predictability or vice versa? Which one is harder to compute?

2. My other concern is with the definition of link predictability. The authors remove 1 link at a time and use existing link prediction algorithms to rank the removed link with respect to non-links. This is a very narrow definition of the link prediction task which may not reflect the full complexity of the problem.

Imagine, you have a perfect square lattice network, where you remove 1 link at a time. It is straightforward to guess each link that is missing. If you remove 10% or, say, 50% of all links at the same time, the problem of link prediction is more challenging.

My question, both as a reader and a referee, is whether the definition of link predictability proposed by the authors is related to the one used in the standard classification literature, be it AUC, AUPR, Precision, F1 score or anything related?

3. I am not sure I fully agree with the authors definition of the network dataset value in Eq. (12), which implies that predictable networks have higher commercial value. Naively, I would think that deterministic networks have little value, as they can be predicted easily. Conversely, I am inclined to think that values of random networks is higher since one can't be sure where the links are unless one

measures them. I invite authors to extend the discussion of the rationale behind their metric on network commercial value.

On the technical level:

4. I note that the authors use the natural logarithm (\ln) and the logarithm of arbitrary base (\log) interchangeably. I invite them to make the presentation consistent by using of the two.

5. Eqs. (7), (8), and (9) imply that the slope of the linear dependence between H and L is dependent on the average degree of the network. At the same time, all the linear dependences between H and L in all synthetic experiments are characterized by the slope independent of $\langle k \rangle$. Have real networks been picked such as their average degrees are the same?

6. The authors hypothesize that the observed differences between the slopes of real networks and the slopes of synthetic networks are due to the fact that real networks have "additional constraints or mechanisms" that are not captured in synthetic models.

To test if this is the case, I invite authors to consider not only randomized/reconstructed real networks but also the genuine edge-independent synthetic networks. One possibility here is to use a degree-corrected stochastic block model or a latent-geometric network model, where nodes are points in the latent space, and node pairs are connected independently with probabilities which are functions of distances in the latent space. Both models have been shown to reproduce structural properties of real networks, beyond degree distributions.

7. Related to comment 2, the link predictability measure of $p_{\{1\}}$ seems rather artificial. I am curious if it is related to any of the standard link prediction scores mentioned above?

8. I believe, there is a typo in the left-hand-side of Eq. (11).

In summary, I do find the linear relationship between the compressibility length and network predictability non-trivial. Yet, I do not see the significant potential impact of this result that would justify the publication of the manuscript in Nature Communications. I invite authors to better highlight the importance of their findings in the revised manuscript, which I will be happy to re-evaluate.

Reviewer #2 (Remarks to the Author):

In Ref [23], the concept of link predictability was first proposed. An indicator, called structural consistency was given based on the first-order perturbation on network adjacency matrix. However, the authors cannot give the prediction limit, and the method involved a parameter which is the size of perturbation set, although the results are not affected by it. I am so happy to see that Sun et al. go one step further in this direction and gave some impressive results. They analytical gave the upper and lower bound estimations, and use it to predict the actual value of a network.

They encoded the network into a binary string, and used network compression algorithms to obtain the shortest compression length possible. They found a robust universal relation between this length and the predictability of the network, namely the best prediction algorithm among 11 existing algorithms has prediction performance that is linear with the shortest network compression length. They further tested that the linear trend is robust against random link shuffling, link addition or deletion. They also demonstrated analytically this linear relationship using artificial random networks.

Their idea is very interesting, and was validated on both real networks and their random counterparts. The theoretical analysis provides better understanding of network prediction, and has wide application to the field of machine learning and big data.

However, the current version still has some problems need to be addressed before acceptance. Here are my suggested improvements:

- 1, What is the predictability of a network, should be clearly defined.
- 2, What are the differences between, structure predictability, link predictability and the predictability of a network? All were used in the paper, but not clearly defined. If they are the same meaning, then unify the expressions.
- 3, Real network was constructed by randomness and regularities. The authors have tested the relation between randomness and network predictability. Can we consider the shortest compression length a measure of network complexity (the network entropy)? If not, what's the difference?
- 4, The evaluation method that the author applied, namely move only one link and calculate its ranking of the scores, is called leave-one-out method which is an extreme case of K-fold cross-validation, see details in section 2 of ref [2].
- 5, More real network should be tested to validate the method. For the tested 11 networks they were very close to the fitting line, see Fig 2, however there must be some outliers if more networks are tested. What is the meaning of the deviation to the line?
- 6, When deriving p_1 and P_C of prediction accuracies, the authors assume $p_1 \geq p_2 \geq p_3 \dots$. Compared with the best prediction algorithm, the upper bound of P_C does not contradict the prediction algorithm's results, showing the reliability of the assumption. I suggest the authors to add the distribution of $p_1, p_2, p_3 \dots$ in the networks studied with the best prediction algorithm, to directly investigate the validity of the assumption.
- 7, "Universal class" is a very important claim in physics. I suggest the authors move Fig S7 to the main manuscript to explain why the slope 0.7 is a universal class.
- 8, The authors mentioned the isomorphic network. This reminds me that, in a network there may have some links that are structural symmetry, if one link was removed the symmetry will be broken, and thus may very difficult to predict out. Actually, these links are not distinguishable. How to consider this in the predictability measurement?

Overall, this is a paper of novelty and interesting. I'm happy to recommend its acceptance if the authors can well address the above questions.

Rebuttal Letter

We thank the editor and referees for the valuable comments and suggestions, and we have carefully revised our manuscript. Our response are as follows.

Reviewer #1:

The authors report a linear relationship between the shortest compression length of a network and its link predictability. This result is established on a collection of real networks.

Authors support their empirical finding by deriving a linear relationship, albeit with a different slope for the class of edge-independent random graphs. Authors argue that the discovered linear relationship may serve as a tool for the estimation of commercial value of the network dataset.

The problem of link prediction and network reconstruction in general is an important problem in data sciences and is certainly worth investigation. The manuscript is fairly well-written and is accessible for non-specialists in the field.

We wish to thank the referee for an accurate summary of the paper, which highlights his/her expertise on the subject. We are delighted to hear that the referee has grasped the key essence of the paper, appreciates the importance of the paper, and thinks the paper is well-written and accessible to a broad audience. We next offer point-by-point response to each of the thoughtful comments the referee shared with us.

1. My main concern is with respect to the importance of the reported finding. While the discovered linear relationship between the compression length and link predictability does not seem obvious, it is not at all clear what its immediate benefits are.

The authors claim that the discovered relationship may help evaluate the commercial value of the network dataset. While this might be true, one can argue that the compression length itself may serve as such indicator: small length implies that the dataset is regular and easy to predict, on the other hand a long compression length implies that the network dataset is more random and therefore is less predictable. I challenge the authors to think about better-defined applications of their finding.

We thank the referee for motivating us to further explore the immediate benefits of our framework. Here the referee raised a rather insightful comment of whether network compression length can be used by itself to gauge predictability of networks. This is an interesting question, one that we have not discussed adequately in our previous manuscript. Here we carry out additional studies to systematically test this hypothesis. As illustrated in Fig. R1(a), we plot the predictability (normalized algorithm performance entropy) from the best of the 11 algorithms vs. network compression length. We find that, in real networks, there is almost no correlation between these two quantities ($R^2 = 0.04$). Additionally, we carried out the same analysis on artificial networks, which have the same degree distributions as the real ones, finding again a modest correlation at best (Fig. R1(b), $R^2 = 0.14$). Similarly, we also failed to find any meaningful correlation between the raw compression length and the unnormalized

algorithm performance entropy (Fig. R1(c), (d)). These results are in sharp contrast to the clear linear relationship (Fig. R1 (e), (f)) we uncovered in our paper, demonstrating that, without the theoretical intuition on proper normalization, the network compression length itself does not serve as a good indicator of predictability. We wish to thank the referee for this insightful comment which further highlights the contribution of our paper. To this end, we now mention these points in the main text, with detailed analyses included in the SI, section V.

Fig. R1 Network compression length and network predictability. The result on 12 real (a, c) and artificial (b, d) networks how lack of significant correlation between the two quantities without proper normalization, compared with the linear relationship discovered in this work (e, f).

At the same time, the thoughtful comment offered by the referee also prompted us to think further about potential applications of our findings, which is why in this revision, we have spent a considerable time working out the additional benefits of our results, which are summarized as follows:

- 1) Based on the linear relationship between the normalized compression length and predictability, we can, in principle, quantify the performance of any prediction algorithm. Fig. R2 compares the Jaccard algorithm [35] performance entropy with the linear relationship. It can be seen that for any network, the further is the Jaccard algorithm performance entropy from the linear line, the less optimal it is from the best possible prediction results. This quantifiable distance can serve as a benchmark for any practical algorithm in any networks. In other words, the further it is from the linear relationship, the higher the potential of improvement for a new algorithm to achieve. Conversely, if the algorithm's performance entropy lies close to the line, it means that the algorithm is already performing quite well, and further exploration on algorithms is unlikely to yield significant improvement, suggesting diminishing returns in developing new algorithms. We have added these discussions in the main text.
- 2) Based on our estimated upper bound of prediction accuracy (Eqn. 10 in the main text) using the linear relationship, one can quantify the maximal fraction of edges that can be predicted, which was not possible before. Not only can it estimate the maximal performance of an employed algorithm, it can also be used to evaluate the potential value of link prediction before prediction algorithms are used or developed.
- 3) Based on 2), one can estimate the network dataset's commercial values of the link predictability through the size of the compressed network data file. To the best of our knowledge, our work represents among the first attempts to quantify commercial value (e.g., dollars) directly through network data's digital size (e.g., bits). Although our current method only offers a conservative estimate, the limited inquiry into this direction highlights the novelty of our results, making this area particularly fallow for future research.

Fig. R2. Normalized performance entropy using Jaccard algorithm vs. compression length over different real-world networks.

We have revised the manuscript by including the discussions above, and highlighted them in red to make it easy for the referee to track.

Lastly, we wish to emphasize the theoretical contribution of our work. The main virtue of our work lies in the fact that predictability estimation from network compression length is independent from any particular prediction algorithm. As such, it offers a new approach to gauge the goodness of any algorithm, providing guidance for algorithm development and its potential. In science, there is always much value in establishing such theoretical bounds. One prominent example is Shannon's coding theorem in the field of information theory, which defines the limit of algorithm, providing benchmark for communications. Indeed, the theoretical upper bound for practical algorithms defined by Shannon's coding theorems represents a significant progress to the field, which continues to have practical applications up to today. For instance, the international coding standards for 5G communication, LDPC code and Polar code both use Shannon's coding theorems to benchmark performance.

The challenge in understanding link predictability lies in the fact that obtaining the exact formation mechanism of the network is difficult, if not impossible. For example, real networks often have numerous short loops that prevent accurate mathematical descriptions over its complex interactions. Partly for this reason, there have been few related works, and prior analyses like Ref [23], while quite extensive and representing significant contributions to the link prediction literature, are still based on specific prediction algorithms to empirically gauge the predictability of algorithms. The main challenge with prediction-algorithm based predictability analysis is that, different algorithms have different performance on the same network. As shown in Table I in the main text, on metabolic network, the RA algorithm has performance entropy as low as 0.38, while the PA algorithm almost doubles this value at 0.75. Additionally, different networks are hardly comparable for their different sizes and densities, making it difficult to have a unified benchmark to independently quantify predictability. From this perspective, our work aims to help further this line of work by establishing theoretical foundations for link predictability.

Imagine, for instance, that you have an incomplete dataset where only a small fraction of links has been captured. Besides, some of the captured links are false-positives. Can the authors think about the way to estimate the completion cost of the dataset?

We thank the referee for this constructive comment. This is an excellent suggestion, and we have considered it carefully in this revision. Broadly speaking, there are two potential scenarios.

- 1) The observed network data has only a fraction of links. This is similar to the case of random link deletion presented in Fig. 2D in the main text. The predictability of the network will decrease in this case, and correspondingly its commercial value for link prediction will decrease as well.
- 2) The observed network data contain false links. If we assume such false links appear randomly, it becomes similar to the case in Fig. 2B in the main text. The predictability will decrease accordingly, so is its prediction value.

The referee's suggestion prompted us to think further: if there is a small fraction of false links or missing links, is it possible to identify such links using network structural entropy (the shortest compression length)? If so, it would be an additional contribution of our findings. The referee's

comment made us realize that one can leverage link prediction algorithm to identify the false links. For instance, the ranking of an existing link according to an algorithm can be related to how likely this link is a false link: the lower is the rank, the more likely it is false.

To develop such an algorithm using our result, we carry out the following initial study. We randomly add F links (e.g. $F = 100$) as false links to an original network, and also choose another T existing links (e.g. $T = 900$). From the seed set of $F+T=1000$ links, we try to use only the network compression length (structural entropy) as an indicator to identify these 100 false links. We use the following simple method: we remove each one of the 1000 links individually, and calculate the compression length of the modified network. The 100 removed links that lead to the shortest compression length L are chosen to be the most likely false links, as random links lead to the longest compression lengths. In Fig. R3(a)-(b), we see that, despite its simplicity, this method clearly outperforms random guess. Its accuracy is not very high at the moment, but with more refinements and combined with more advanced techniques like simulated annealing, genetic algorithm etc., its accuracy may be further improved.

Similarly, we use the same method to identify missing links. In a network with full information on existing links, we randomly remove $F=100$ links as missing links. At the same time, we chose $T=900$ links that are not present. Every time we add one link from the 1000 links, we calculate the network's compression length. The 100 links that give the shortest compression lengths are then chosen as the missing links. From Fig. R3 (c)-(d), we see that this simple method again outperforms random guess.

The above results imply that, network compression length (structural entropy) can be used as an independent indicator to identify missing or false links. We also appreciate another great point raised by the referee regarding estimating completion cost of the dataset, which highlights another close connection between our results and the broader network literature. While we recognize that this problem may be more closely related to the network reconstruction problem [S. Hempel, A. Koseska, J. Kurths and Z. Nikoloski, *Physical review letters*, 2011.], [Z. Shen, W.-X. Wang, Y. Fan, Z. Di and Y.-C. Lai, *Nature communications*, 2014.] rather than link prediction, we wish to thank the referee for pointing out this connection, and we have now added to the paper to discuss and reference this potential connection.

Fig. R3 Using structural entropy as indicator to identify false-positive links (a)-(b) and missing links (c)-(d).

Can one use the compression length to estimate link predictability or vice versa? Which one is harder to compute?

We thank the referee for this insightful comment. Yes, we can use one to predict the other thanks to the linear relationship uncovered in Eqn. 3 in the main text. The compression length only depends on network topology, and the computational complexity is only $O(N + E)$ [25]. In general, the compression length can be reliably calculated as the field is well established and techniques are well developed. On the other hand, to ensure reliable estimation close to true predictability, it requires implementation of a large number of different algorithms, usually with high computational complexity. Specifically, for calculating the rank distribution entropy (network predictability), we need to employ an algorithm for every link in a network. Consequently, the computational complexities of algorithms like SPM [23] and LRW [33] in estimating H , are about $O(N^3E)$ and $O(N(k)^nE)$, respectively, where n is an integer constant. Therefore, it is more efficient and reliable to use network compression length to estimate predictability. We have revised this point in the manuscript with red color.

2. My other concern is with the definition of link predictability. The authors remove 1 link at a time and use existing link prediction algorithms to rank the removed link with respect to non-links. This is a very narrow definition of the link prediction task which may not reflect the full complexity of the problem.

Imagine, you have a perfect square lattice network, where you remove 1 link at a time. It is straightforward to guess each link that is missing. If you remove 10% or, say, 50% of all links at the same time, the problem of link prediction is more challenging.

My question, both as a reader and a referee, is whether the definition of link predictability proposed by the authors is related to the one used in the standard classification literature, be it AUC, AUPR, Precision, F1 score or anything related?

Here the referee raised another thoughtful comment, which motivated us to think more carefully about our definition, and its relationship with the broader literature. Indeed, the other existing ranking methods [2, 3, 23] which pre-remove x fraction of links (x is usually 5% or 10%) are similar to the leave-one-out method [31, 32] used in our work, and would yield similar results as leave-one out if x is very close to 0. Specifically, these existing methods first remove x fraction of links and use the remaining links to calculate a score for the links not present in the remaining network. While the leave-one-out method only removes 1 link at a time and then calculate its ranking value. For instance, for a network with 1000 links, we need to calculate the ranking 1000 times. For other existing methods, every time x fraction of links is removed, the ranking only needs to be done once for the removed links together. Hence if x is chosen to be very close to 0, and the distribution entropy is measured by repeatedly sampling and taking the average, it becomes similar to our method as well. Effectively, this is approximately a sub sample in our 1000 sample points. Qualitatively, doing so would yield similar results as our method by removing 1 link at a time as shown in Fig. R4 below. It shows that on the Metabolic and USAir networks using BPAA algorithm, our method has the similar ranking distribution and its entropy value with the repeated sampling for removing x fraction of links. As the fraction of removed links decreases from 10% to 1%, the result is approaching ours. However, without repeated sample on removing x fraction, the fluctuation in distribution is much larger as shown in Fig. R5.

By removing only one link at a time in our method, there are two major advantages - 1. Compared to removing x fraction of links, our method is parameter-free. That is, we do not need to consider the impact of x . 2. Doing so preserves the original network structure as much as possible, such that the intrinsic predictability is also preserved. Even though it is possible to remove more than 1 link at a time, removing different batches of links would result in different performance entropy values as shown in Fig. R6 (a)-(c) and (e)-(g). Such fluctuation would result in uncertainty in the results without clear advantage. On the contrary, our method has minimal fluctuation in the results as shown in Fig. R6 (d) and (h), by preserving the original network as much as possible. This facilitates us to find more accurate linear relationship between the network compression length and predictability. drawback is the higher computational complexity compared with only using 1 sample of removing x fraction of links. But if multiple samples of removing x fraction is used, the advantage in computational complexity of those popular methods will not be significant over ours.

Using the performance entropy or network predictability we could obtain the upper and lower bounds of Precision. You can see the response of comment 7. **Here, we would like to point out that using rank distribution entropy to measure the algorithm performance is the innovation of our work.** We have revised the main text accordingly in red and have included the discussion above in the SI, section IV.

Fig. R4. The relation of the leave-one-out method and other methods on our newly-proposed algorithm performance entropy. For other methods, we used multiple samples of removing x fraction of links to obtain the average behaviors. As the fraction of removal decreases (10%-5%-1%), the entropy value decreases and approaches our method for removing 1 link at a time.

Fig. R5. Similar to the Fig. R4, but here we only use 1 sample of removing x fraction. The fluctuation in ranking values is significant, and more pronounced for small x .

Fig. R6. Algorithm performance entropy in Metabolic network obtained by different methods. (a)-(d) RA algorithm, which is a deterministic algorithm. (e)-(h) SPM algorithm, which involves randomness.

3. I am not sure I fully agree with the authors definition of the network dataset value in Eq. (12), which implies that predictable networks have higher commercial value. Naively, I would think that deterministic networks have little value, as they can be predicted easily. Conversely, I am inclined to think that values of random networks is higher since one can't be sure where the links are unless one measures them. I invite authors to extend the discussion of the rationale behind their metric on network commercial value.

We fully agree with the referee that the intrinsic value of a network dataset is minimal if it is deterministic and predictable, as the buyer of the data does not need to buy the data to know the structure inside. Our work focuses on the derived value of the networks: that is, the possible new links that can be predicted using the existing network dataset. In other words, we focus on the value of the potential or unobserved links using existing links, not existing links themselves. This could be the case for drug discovery, for instance, that one wants to predict new possible drugs using existing data; or recommendation systems for products that the particular consumer has never bought before. But since these two concepts of values are highly related, the referee's comment motivates us to think deeper about the value of network dataset.

In practice, when someone intends to pay for a network dataset, the price or value is related to how much additional commercial benefit can be retrieved from it. In general, there can be two situations: First, if there is no alternative information, and the buyer uses only the network data and develop prediction algorithms to benefit from the data, and say the amount of benefit is V ; Second, if there are other information available that helps the buyer understand the network data even without having the data, the buyer can still have a baseline benefit from such external information, say Z amount. The later could be the case when a movie producer has some experience about the overall market without knowing who watches what types of movies. For instance, a movie producer may achieve available prediction results by simply recommending well-known popular movies with good box office to the audience. So, if the value Z is captured within V , the additional benefit from the dataset combined with prediction algorithm is $V - Z$.

As mentioned by the referee, deterministic network like the lattice does not need much information to know its exact structure and carries less value. For random networks like the ER network, it is not possible to know any better than random guess for predicting links, and the value of the existing links is high, but the predictive value is low. Hence, we can assume that the value Z and $f(V)$ are related, say $Z = f(V)$. Next, we could quantify the additional value of link prediction, $W = V - f(V)$ (Fig. R7(a)). There are different scenarios: (1) For $f(V) = 0$, corresponding to the case discussed in the main text. (2) For regular structure like the lattice, $f(V) = V$ as the network structure is fully captured by other known information. (3) For totally random network like the ER network, both $V = 0$ and $f(V) = 0$.

We next examine the 3 cases separately, as illustrated in Fig. R7(b) and (c). The real-world case should be the mixture of the following three cases.

- (1) For the case of $f(V) = 0$ represented by the red line, there is no external information, or such information does not overlap with the predictive benefit from the dataset. The additional value

of the dataset is simply $W = V$.

- (2) For $f(V) = V$ in green, external information (i.e. knowing that it is lattice structure) leads to the value V equal to Z , when external information fully captures the network data. In such a case $W = 0$ and no additional value is provided by the dataset.
- (3) $f(V)$ increases together with V but smaller than V , shown in blue. Here the additional value from the dataset $W = V - f(V)$ is significant and corresponds to many real problems when different information/dataset contributes to the total value.

The commercial value we refer to here is the additional value that can be realized by predicting unobserved information from the dataset. For example, in life science research, obtaining the protein-protein interactions needs extensive experimentations that cost both time and resources; if one can make predictions on such interactions to narrow the experimentation scope, it can reduce the cost of experiments, which is the predictive value of the existing dataset. A way to quantify such value is the cost per experiment multiplied by the reduction in the number of experiments to obtain accurate results. As we are not concerned with or able to know the base value Z or $f(V)$ from information outside the network dataset, we focus only on the predictive value of the dataset. We have included this discussion in the manuscript in red and have added detailed discussion in the SI, section X.

Fig. R7. Value of the network dataset considering both its predictive value and its baseline value.

On the technical level:

4. I note that the authors use the natural logarithm (ln) and the logarithm of arbitrary base (log) interchangeably. I invite them to make the presentation consistent by using of the two.

Thanks for the suggestion. \ln and \log denote the logarithmic operation with base e and 2 respectively. We have revised them in the manuscript (in red).

5. Eqs. (7), (8), and (9) imply that the slope of the linear dependence between H and L is dependent on the average degree of the network. At the same time, all the linear dependences between H and L in all synthetic experiments are characterized by the slope independent of $\langle k \rangle$. Have real networks been picked such as their average degrees are the same?

We thank the referee for this comment. The real networks have different average degrees. The slope of the linear relationship is related to both $\langle k \rangle$ and N . As seen in Fig. 3D in the revised main text, the two terms $2/\langle k \rangle$ and $\log\langle k \rangle/\log N$ compete with each other in determining the slope, resulting in similar values in slope. We have added the average degree and size of the real networks into Table 1 for a clearer presentation.

6. The authors hypothesize that the observed differences between the slopes of real networks and the slopes of synthetic networks are due to the fact that real networks have "additional constraints or mechanisms" that are not captured in synthetic models.

To test if this is the case, I invite authors to consider not only randomized/reconstructed real networks but also the genuine edge-independent synthetic networks. One possibility here is to use a degree-corrected stochastic block model or a latent-geometric network model, where nodes are points in the latent space, and node pairs are connected independently with probabilities which are functions of distances in the latent space. Both models have been shown to reproduce structural properties of real networks, beyond degree distributions.

We thank the referee for this very insightful comment that led us to explore deeper into the drivers behind the observed slope. We agree that it is a great idea to use "genuine edge-independent synthetic networks" to see if they help explain the value of the empirical slope, and understand better the difference between the empirical and theoretical values.

We have implemented two models suggested by the referee. The first degree-correlated stochastic block model [Karrer & Newman, *Physical review E*, 2011] is a generative model for community structures. According to the original paper, node i and j has probability $\theta_i\theta_j\omega_{g_i g_j}$ of being connected, where ω_{rs} is a symmetric matrix of parameters controlling edges between groups r and s , g_i is the group assignment of vertex i , and θ_i is set of parameters controlling the expected degrees of vertices i . We assume the number of communities is 2, apply degree distributions from empirical networks, and use Eqns. 12, 18 and 27 in [Karrer & Newman, *Physical review E*, 2011] to determine the values of ω, θ, g . With this we generated an artificial network with 2 communities and same degree distribution as the empirical networks; the community strength can be controlled with parameter λ : when $\lambda=1$ the links are only within each community; when $\lambda=0$ the links are random but degrees are preserved similar to our treatment in the main text. After determining the values of ω, θ, g , we obtain the adjacency matrix Q , and use the method in our work to calculate L^* and $H_{\text{TBP A}}^*$. The results are

shown in the Fig. R8(a) below. We observe that the degree-corrected stochastic block model still falls on the theoretical linear relationship that is different from the empirical one. In addition, as λ decreases, the community structure weakens, the position of (L^*, H_{TBPA}^*) in the figure shifts upwards, in line with our theoretical results.

For the second latent-geometric network model [Newman & T. P. Peixoto, *Physical review letters*, 2015][Bernard M. Waxman, *IEEE journal on selected areas in communications*, 1988], link probability depends on the latent geometric distance between the two nodes. Here we consider the latent geometric space as 1-D real axis \mathbb{R}^1 , and the nodes are randomly distributed on the line.

According to [Newman & T. P. Peixoto, *Physical review letters*, 2015], q_{ij} equals to $\frac{k_i k_j}{2m} \delta(i, j)$, where k_i, k_j are the degrees of the nodes, m is the total number of edges, and $\delta(i, j)$ is the edge function. For $\delta(i, j)$, without loss of generality, we consider the following case: the probability of connecting two nodes is inversely related to their distance. Hence, we employ the method in [Bernard M. Waxman, *IEEE journal on selected areas in communications*, 1988], and set $\delta(i, j) = \beta \exp\left(\frac{-d(i, j)}{\alpha L}\right)$, where $d(i, j)$ is the distance between i and j , and L is the maximum distance between two nodes, $\alpha \in (0, 1]$ controls the strength of the geometric influence: larger α leads to more random connections. While parameter β controls the average degree of the network, ensuring the artificial network is similar in total edges. Next, we use the adjacency matrix Q to calculate L^* and H_{TBPA}^* . The results are shown in Fig. R8(b) below. Again, the values fall on the theoretical linear relationship, and the point (L^*, H_{TBPA}^*) shifts towards top right as the geometric influence weakens.

Fig. R8. Normalized TBPA performance entropy vs. compression length for (a) degree-correlated stochastic block model and (b) latent space model. Both follow the same linear relationship as our theoretical result.

In our theoretical derivation of the linear relationship, both L^* and H^* are based on the matrix Q that generates the artificial networks, with element q_{ij} denoting the probability of connecting node i and j . Importantly, we did not enforce any particular form of Q . Hence, for the above two models,

although they have more complex and different Q s, the predictability properties are still within the analytical scope of our theory. Specifically, as long as the edges are independently formed and their probabilities can be described using adjacency matrix Q , the network's predictability-compression length relation should follow our theoretical results in Eq. (7). We have included the content above in the SI, section VIII.

7. Related to comment 2, the link predictability measure of $p_{\{1\}}$ seems rather artificial. I am curious if it is related to any of the standard link prediction scores mentioned above?

We thank the referee for pointing this out. The predictability measure is designed to provide a measure that is independent of the network size N , which is also related to the standard measure like precision. According to Ref [2], if there are Tr number of links predicted correctly among the top T predictions, the precision is Tr/T . In our work, p_1 refers to the fraction of correct links among the top N predicted links, and p_c refers to the fraction among the CN top predicted links. Hence if we take $T = N$, then p_1 is equivalent to precision; if $T = CN$, then p_c is the precision. We have revised this point in the manuscript.

8. I believe, there is a typo in the left-hand-side of Eq. (11).

We failed to identify the typo mentioned in the equation, but would be delighted to examine further if needed.

In summary, I do find the linear relationship between the compressibility length and network predictability non-trivial. Yet, I do not see the significant potential impact of this result that would justify the publication of the manuscript in Nature Communications. I invite authors to better highlight the importance of their findings in the revised manuscript, which I will be happy to re-evaluate.

We would like to thank referee for the very insightful and constructive comments. They have inspired many new analyses that we believe have led to a stronger paper.

Overall, we believe that the most significant contribution of this work is: for the first time, we provide an entropy-based measure of the predictability of complex network structure that is independent of any algorithm, and a specific theoretical derivation on random artificial networks. Secondly, this measure of predictability and its linear relationship can be used to estimate the gap in performance between an existing algorithm and the prediction limit, which is theoretically demonstrated on random artificial networks and empirically validated on real networks. This will provide a new benchmark for link prediction algorithm. Besides, our theory allows the estimation of the commercial value based on the predictability according to the normalized size of compressed network data. To the best of our knowledge, this is among the first time that a direct relationship between data size and commercial value (bit to dollar) has been established.

Please do not hesitate to let us know if there is anything further we can do to improve this work.

Reviewer #2

In Ref [23], the concept of link predictability was first proposed. An indicator, called structural consistency was given based on the first-order perturbation on network adjacency matrix. However, the authors cannot give the prediction limit, and the method involved a parameter which is the size of perturbation set, although the results are not affected by it. I am so happy to see that Sun et al. go one step further in this direction and gave some impressive results. They analytical gave the upper and lower bound estimations, and use it to predict the actual value of a network.

They encoded the network into a binary string, and used network compression algorithms to obtain the shortest compression length possible. They found a robust universal relation between this length and the predictability of the network, namely the best prediction algorithm among 11 existing algorithms has prediction performance that is linear with the shortest network compression length. They further tested that the linear trend is robust against random link shuffling, link addition or deletion. They also demonstrated analytically this linear relationship using artificial random networks. Their idea is very interesting, and was validated on both real networks and their random counterparts. The theoretical analysis provides better understanding of network prediction, and has wide application to the field of machine learning and big data.

However, the current version still has some problems need to be addressed before acceptance. Here are my suggested improvements:

1, What is the predictability of a network, should be clearly defined.

We thank the referee for pointing this out. We define network predictability as the best possible performance for any link prediction algorithm, which is measure by entropy. We have revised the manuscript accordingly in red.

2, What are the differences between, structure predictability, link predictability and the predictability of a network? All were used in the paper, but not clearly defined. If they are the same meaning, then unify the expressions.

Thanks for this comment. In our manuscript, these three phrases were used in an interchangeable manner. Following the referee's comment, we have unified them using "structure predictability" in this revision.

3, Real network was constructed by randomness and regularities. The authors have tested the relation between randomness and network predictability. Can we consider the shortest compression length a measure of network complexity (the network entropy)? If not, what's the difference?

We thank the referee for this comment. According the Shannon source coding theorem, the shortest possible compression length of the network measures its entropy (detailed proof in Ref [25]), which

is related to H . This means, given a network formation mechanism, the logarithmic value (with base 2) of the number of typical network structure (isomorphic networks are considered as 1) is the entropy value.

4, The evaluation method that the author applied, namely move only one link and calculate its ranking of the scores, is called leave-one-out method which is an extreme case of K-fold cross-validation, see details in section 2 of ref [2].

We thank the referee for pointing this out, which highlights the referee’s deep knowledge about the field. We have revised our manuscript by referencing the ‘leave-one-out’ method in Ref [2].

5, More real network should be tested to validate the method. For the tested 11 networks they were very close to the fitting line, see Fig 2, however there must be some outliers if more networks are tested. What is the meaning of the deviation to the line?

We thank the referee for this valuable comment. Indeed, it is possible that real networks may deviate from the straight line predicted by our framework, as the compression length obtained by the employed compression algorithm is an approximation, hence not strictly equal to the real structural entropy of the network. To this end, we have taken the referee’s thoughtful advice and collected data on 6 new networks spanning social, biological and infrastructural domains. There are in total 18 different networks including the previous 12 networks now in the revised figure. As shown in Fig. R9, the new networks have the same linear relationship as the previous 12. We see the addition of the 6 datasets as a strong support and validation of our existing findings, and wish to thank the referee for the great suggestion.

Fig. R9. Normalized BPAA performance entropy vs. normalized compression length with 18 empirical network datasets.

The newly added networks include: EuroRoad [Šubelj, L. & Bajec, M. Robust, 2011] is a European

road network, Physicians [James Coleman, Elihu Katz, & Herbert Menzel, 1957] is collaboration network among physicists, Rt-twitter [Ryan A. Rossi and Nesreen K. Ahmed, 2015] is a subnetwork on Twitter social network, and Bio-diseasome, Bio-CE-LC and Bio-DM-LC [Ryan A. Rossi and Nesreen K. Ahmed, 2015] are three different protein interaction networks.

6, When deriving p_1 and P_C of prediction accuracies, the authors assume $p_1 \geq p_2 \geq p_3 \dots$. Compared with the best prediction algorithm, the upper bound of P_C does not contradict the prediction algorithm's results, showing the reliability of the assumption. I suggest the authors to add the distribution of $p_1, p_2, p_3 \dots$ in the networks studied with the best prediction algorithm, to directly investigate the validity of the assumption.

Thanks for the suggestion. The distribution of p_j from the best performing algorithms of all 12 networks are in following Fig.R10. We also put this figure in the revised SI (Fig. S25).

Fig.R10. Distribution of p_j in the 12 networks studied with the BCAA.

7,"Universal class" is a very important claim in physics. I suggest the authors move Fig S7 to the main manuscript to explain why the slope 0.7 is a universal class.

Thanks for the suggestion. We have moved Fig. S7 to the main text, and have added corresponding discussion to explain it.

8, The authors mentioned the isomorphic network. This reminds me that, in a network there may have some links that are structural symmetry, if one link was removed the symmetry will be broken, and thus may very difficult to predict out. Actually, these links are not distinguishable. How to consider this in the predictability measurement?

We thank the referee for another great comment. We agree that we should have discussed structural symmetry as well in the paper. As discussed in detail in Ref [25], the probability for a network to have symmetrical structure G is upper bounded by $O(N^{-\omega})$, for arbitrarily small positive value ω . Since N is the size of the network, which tends to be very large, this possibility can approach to zero in large N limit, which may explain why it is difficult to have structural symmetry in real networks. This is consistent with the networks studied in our paper, as many of them can be driven by formation mechanisms that are largely stochastic. We have now added these discussions to the SI, section VIII.

Overall, this is a paper of novelty and interesting. I'm happy to recommend its acceptance if the authors can well address the above questions.

We would like to thank the referee for the his/her very constructive comments and positive opinions of our work!

REVIEWERS' COMMENTS:

Reviewer #1 (Remarks to the Author):

My concerns have been addressed in the revision and now I can recommend the manuscript for publication.

Reviewer #2 (Remarks to the Author):

The authors have well addressed both referees' comments and justified well the applications of their achievements. And I found that their method can be considered as a new benchmark for testing the absolute performance of link prediction algorithms. I appreciate the authors' insights on this topic and recommend its acceptance.